# Structural insights for selective disruption of Beclin 1 binding to Bcl-2

Yun-Zu Pan[1,2,3], Qiren Liang[2], Diana R. Tomchick [1], Jef K. De Brabander [2] & Josep Rizo [1,2,3 ✉]

Stimulation of autophagy could provide powerful therapies for multiple diseases, including cancer and neurodegeneration. An attractive drug target for this purpose is Bcl-2, which inhibits autophagy by binding to the Beclin 1 BH3-domain. However, compounds that preclude Beclin 1/Bcl-2 binding might also induce apoptosis, which is inhibited by binding of Bcl-2 to BH3-domains of pro-apoptosis factors such as Bax. Here we describe the NMR structure of Bcl-2 bound to **35**, a compound that we recently found to inhibit Beclin 1/Bcl-2 binding more potently than Bax/Bcl-2 binding. The structure shows that **35** binds at one end of the BH3-binding groove of Bcl-2. Interestingly, much of the **35**-binding site is not involved in binding to Bcl-2 inhibitors described previously and mediates binding to Beclin 1 but not Bax. The structure suggests potential avenues to design compounds that disrupt Beclin 1/Bcl-2 binding and stimulate autophagy without inducing apoptosis.

[1] Department of Biophysics, University of Texas Southwestern Medical Center, Dallas, TX 75390, USA. [2] Department of Biochemistry, University of Texas Southwestern Medical Center, Dallas, TX 75390, USA. [3] Department of Pharmacology, University of Texas Southwestern Medical Center, Dallas, TX 75390, USA. ✉email: Jose.Rizo-Rey@UTSouthwestern.edu

Macroautophagy (below referred to as autophagy) constitutes the major catabolic mechanism to eliminate unwanted cellular components such as damaged organelles, protein aggregates or invading microorganisms in eukaryotic cells. This process involves the formation of double-membrane structures called autophagosomes that are targeted to lysosomes for degradation of their contents and is fundamental for multiple physiological functions, including maintenance of nutrient homeostasis, control of organelle quality, development, cell differentiation and defense against microbes among others[1–4]. Correspondingly, mutations in genes that control autophagy have been linked to a variety of diseases such cardiovascular, neurodegenerative, metabolic, pulmonary, renal, infectious and musculoskeletal disorders, as well as diabetes and cancer[5,6]. These observations correlate with the phenotypes observed upon genetic disruption of autophagic genes[2,5,6]. Conversely, the life span and health of various animal models are enhanced in the absence of proteins that suppress autophagy such as Bcl-2 and Rubicon, which stimulates basal autophagy in vivo[7–10]. Hence, strategies that enhance autophagy have the potential to provide novel therapies for varied human diseases. Unfortunately, no drugs that directly and specifically activate autophagy have been described.

A promising target for drug design directed at enhancing autophagy is Bcl-2, which binds to the essential autophagy factor Beclin 1[2]. This protein provides a key scaffold to form class III phosphatidylinositol 3 kinase (PI3KC3) complexes that are critical for initiation of autophagosome formation as well as maturation[11]. Formation of these complexes is prevented under basal conditions by binding of Beclin 1 to Bcl-2 through its BH3 domain, and is activated by diverse molecular events that release the Beclin 1/Bcl-2 interaction, including phosphorylation of a long flexible loop of Bcl-2 or of the N-terminal region of Beclin 1 containing the BH3 domain[11]. The potential of disruption of the Beclin 1/Bcl-2 interaction for therapeutic use was highlighted by the phenotypes observed in knockin mice bearing an F121A mutation in the BH3 domain of Beclin 1 that disrupts Bcl-2 binding and leads to enhanced autophagy in diverse tissues[8]. Thus, this mutation led to increased life span in both males and females, together with decreased propensity to tumorigenesis and to age-related pathological changes in heart and kidney[8].

Most of Bcl-2 forms a globular cytoplasmic domain spanning residues 1-206[12], which is followed by a juxtamembrane sequence (residues 207-218) and a C-terminal transmembrane (TM) sequence. Structural studies of Bcl-2 and the closely related homologue Bcl-xL by NMR spectroscopy and X-ray crystallography have shown that their globular domain forms multiple α-helices and contains a hydrophobic groove that binds to BH3 domains[12–16]. These sequences of about 25 amino acid residues form an α-helix upon binding to members of the Bcl-2 family and are present in diverse proteins, particularly pro-apoptotic proteins such as Bax and Bak[17]. In fact, Bcl-2 acts an apoptosis inhibitor by binding to these proteins and, as expected, the binding modes of their BH3 domains to Bcl-2 or Bcl-xL are similar to those observed for the Beclin 1 BH3 domain[13–16]. Thus, a potential drawback of compounds that interfere with Beclin 1/Bcl-2 binding is that they might activate not only autophagy but also apoptosis. Indeed, organic compounds that mimic BH3 domains and bind to Bcl-2, such as ABT-737, stimulate both autophagy and apoptosis[18].

In a structure-activity relationship study designed to optimize a compound that was previously found to inhibit the Beclin 1/Bcl-2 interaction in a high-throughput screen[19], we recently discovered several compounds that potentially and selectively prevent this interaction[20]. Compound 35 identified in this study (Fig. 1) was particularly potent in inhibiting binding of the Beclin 1 BH3 domain to the cytoplasmic region of Bcl-2 (residues 1-218) in an AlphaLISA assay (IC50 4.4 nM) while exhibiting a much weaker activity in suppression of Bax BH3/Bcl-2 binding (IC50 0.88 μM; 200-fold selectivity)[20]. This selectivity may arise in part because Bax BH3 has a considerably higher affinity than Beclin 1 BH3 for Bcl-2[15,16], but ABT-737 was only 20-fold more potent in inhibiting binding of the Beclin 1 BH3 peptide to Bcl-2 than binding of the Bax BH3 peptide[20]. Hence, it is plausible that 35 disrupts specific interactions of Bcl-2 with residues of the Beclin 1 BH3 domain that are not conserved in the Bax BH3 sequence.

Here we describe NMR studies in solution that were designed to test this hypothesis and elucidate how 35 binds to Bcl-2. We show that the juxtamembrane sequence formed by residues 207-218 is critical for tight binding of 35 to Bcl-2. Structural studies by NMR spectroscopy and X-ray crystallography were hindered by the poor solubility of 35 and the limited stability of the Bcl-2(1-218)/35 complex. However, using NMR spectroscopy, we were able to elucidate the solution structure of a complex of 35 with a chimera in which the long flexible loop of Bcl-2(1-218) was replaced by a shorter sequence from the homologous loop of Bcl-xL (below referred to as Bcl-2-xL). The structure shows how 35 is mostly buried inside Bcl-2-xL, interacting on one side with the juxtamembrane region, which acts as a lid, and on the other side with a surface at one end of the BH3-binding groove. Most of this surface is not involved in binding to other Bcl-2 inhibitors described previously, including ABT-737, and part of this surface is involved in interactions with a few residues at the N terminus of the Beclin 1 BH3 domain but not to the Bax BH3 domain[15]. Our results suggest that optimization of compounds that interact with this area of Bcl-2 may provide an avenue to develop drugs that selectively stimulate autophagy without inducing apoptosis, and yield a structural framework to pursue this goal.

## Results

**The juxtamembrane region is required for tight binding of Bcl-2 to compound 35.** The AlphaLISA assays that showed the tight, selective binding of 35 to Bcl-2 were performed with a fragment spanning the entire cytoplasmic region of Bcl-2, including the juxtamembrane region[20]. In general, Bcl-2 fragments that lack the juxtamembrane sequence were normally sufficient for binding to BH3 domains or to Bcl-2 inhibitors described previously, and such fragments were used for the structural studies that revealed the corresponding binding modes (e.g. refs. [12,15,16,21]). To investigate how the juxtamembrane region influences the binding to Bcl-2 to 35, we acquired transverse relaxation optimized (TROSY) $^1$H-$^{15}$N heteronuclear single quantum coherence (HSQC) of a Bcl-2 fragment that spanned residues 1-206 and hence lacked the juxtamembrane region [Bcl-2(1-206)], and compared them with spectra obtained for Bcl-2(1-218) under analogous conditions (Fig. 2). We observed marked differences in the spectra of Bcl-2(1-206) and Bcl-2(1-218) (Fig. 2a), showing that the juxtamembrane region interacts intramolecularly with the globular domain. Since the sequence of the juxtamembrane region (RPLFDFSWLSLK) is rather hydrophobic, this intramolecular interaction likely involves the hydrophobic groove that normally binds to BH3 domains. As described previously[20], addition of 35 caused extensive changes in the $^1$H-$^{15}$N TROSY HSQC spectrum of Bcl-2(1-218) (Fig. S1a). In contrast, 35 did not produce practically any perturbations in the spectrum of Bcl-2(1-206) (Fig. 2b). These results need to be interpreted with caution because 35 has very limited solubility in water and, to observe binding of 35 to Bcl-2(1-218), the compound had to be added in large excess to diluted protein (1 μM), followed by concentration of the sample to acquire $^1$H-$^{15}$N TROSY HSQC spectra[20]. Hence, it is plausible that 35 binds weakly to Bcl-2(1-2-6) but it is difficult to observe such binding because of weak

**Fig. 1 Chemical structure of 35.** The chiral center (indicated by a *) of the enantiomer that binds to Bcl-2 has the S configuration based on the intermolecular NOE data. The nomenclature used for hydrogen atoms in structure calculations is indicated by the red labels.

affinity and the insolubility of the compound. Regardless of this possibility, it is clear that tight binding of Bcl-2 to **35** requires the juxtamembrane region.

We also investigated whether the juxtamembrane region influences the binding of Bcl-2 to its potent inhibitor ABT-737 or to a peptide spanning the Beclin 1 BH3 domain. Addition of ABT-737 also caused extensive changes on the $^1$H-$^{15}$N TROSY-HSQC spectrum of Bcl-2(1-218) (Fig. 2c), but the changes were drastically different from those caused by **35** (Fig. S1b), indicating that there are important differences in the binding modes of the two compounds. ABT-737 also caused drastic changes in the $^1$H-$^{15}$N TROSY-HSQC spectrum of Bcl-2(1-206) (Fig. S1c), and the spectra of the ABT-737 complexes of both fragments were very similar except for the presence of a few additional cross-peaks in the spectrum of the ABT-737/Bcl-2(1-218) complex that must arise from the juxtamembrane sequence (Fig. 2d). These results show that the binding modes of ABT-737 to both Bcl-2 fragments are analogous. Similar results were obtained in the presence of the Beclin 1 BH3 peptide, which induced changes in the $^1$H-$^{15}$N TROSY-HSQC spectrum of Bcl-2(1-218) that were clearly different from those caused by **35** (Figs. 2e, S1d) and also bound to Bcl-2(1-206) (Fig. S1e), yielding a $^1$H-$^{15}$N TROSY-HSQC spectrum that was superimposable with that of Bcl-2(1-218) except for a few additional peaks in the BH3/Bcl-2(1-218) spectrum (Fig. 2f). Hence, the Beclin 1 BH3 domain binds to both fragments in the same mode. It is noteworthy that the extra cross-peaks observed for the complexes of Bcl-2(1-218) bound to ABT-737 or Beclin1 BH3 are in similar positions (Fig. 2d, f). One of these extra cross-peaks is in the lower left corner of the spectrum where aromatic Trp NH groups normally appear, consistent with the presence of a Trp residue in the juxtamembrane sequence, and the others exhibit poor dispersion in the $^1$H dimension, which is characteristic of unstructured polypeptides. These results show that binding of ABT-737 or Beclin 1 to Bcl-2(1-218) releases the intramolecular

interactions of the juxtamembrane region with the globular domain, supporting the notion that this region binds to the hydrophobic BH3-binding groove of the globular domain. These findings contrast with the observation that the juxtamembrane region is required for tight binding of **35** to Bcl-2 (Figs. 2b, S1a). Nevertheless, it is plausible that **35** also binds to the same groove as ABT-737 and BH3 domains, and displaces the juxtamembrane region, but the displaced juxtamembrane region interacts with **35** and contributes to the overall affinity (see below).

**Structure of the Bcl-2-xL/35 complex.** To investigate how the juxtamembrane region interacts intramolecularly with the globular domain of Bcl-2 and how Bcl-2 recognizes **35**, we performed extensive crystallization trials of Bcl-2(1-218) free and bound to **35** (see "Methods"), but no promising leads were observed. It is worth noting that structural studies of Bcl-2 by NMR and X-ray crystallography were previously shown to be hindered by the long flexible loop, but the behavior of the protein was improved by replacing this loop with a short sequence from the homologous loop of Bcl-xL, which allowed structure determination in solution by NMR spectroscopy[12] and has facilitated a plethora of subsequent structural studies of Bcl-2 bound to diverse ligands. Hence, we prepared an analogous chimera in which the long flexible loop of Bcl-2(1-218) was replaced by the same Bcl-xL loop sequence used in other studies. We refer to the resulting fragment as Bcl-2-xL. The $^1$H-$^{15}$N TROSY-HSQC spectrum of Bcl-2-xL was similar to that of Bcl-2(1-218), with natural differences arising from the distinct loops of the two fragments, and exhibited similar shifts upon binding to **35** (Fig. S2). Hence, the loop exchange did not affect the **35** binding mode.

We also performed extensive crystallization trials with Bcl-2-xL with or without bound **35**, but the trials were again unsuccessful. Structure determination of isolated Bcl-2-xL by NMR spectroscopy was hindered by its propensity to aggregate at

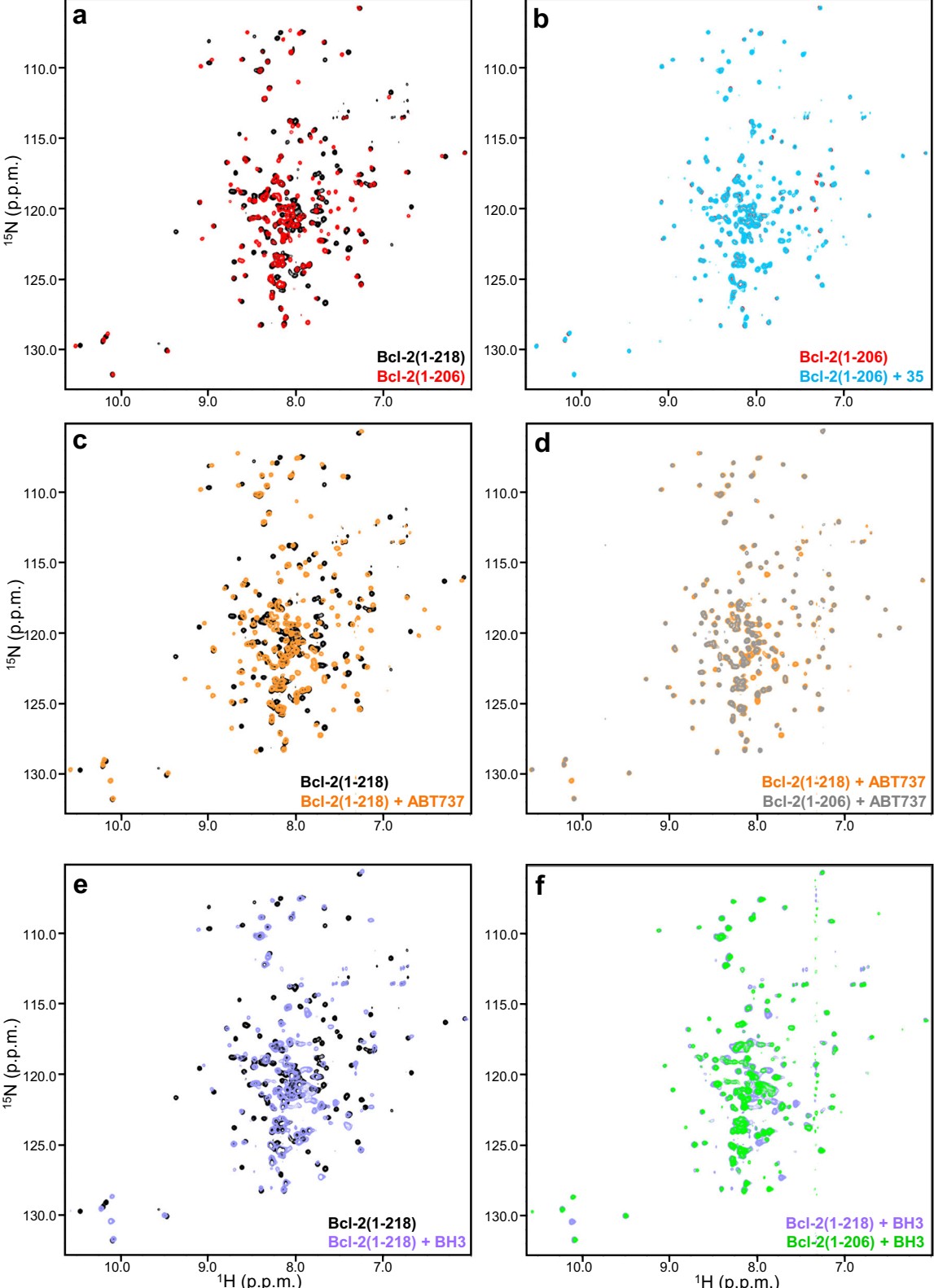

**Fig. 2 Comparison of ¹H-¹⁵N TROSY-HSQC spectra of Bcl-2(1-206) and Bcl-2(1-218) alone and in the presence of different ligands.** The contour plots show superpositions of ¹H-¹⁵N TROSY-HSQC spectra of: (**a**) isolated Bcl-2(1-206) and Becl-2(1-218); (**b**) Bcl-2(1-206) in the absence and presence of **35**; (**c**) Bcl-2(1-218) alone and in the presence of ABT-737; (**d**) Bcl-2(1-206) and Bcl-2(1-218) in the presence of ABT-737; (**e**) Bcl-2(1-218) alone and in the presence of Beclin 1 BH3 domain; and (**f**) Bcl-2(1-206) and Bcl-2(1-218) in the presence of Beclin 1 BH3 domain. The contours plots are color coded as indicated by the labels at the bottom right corner.

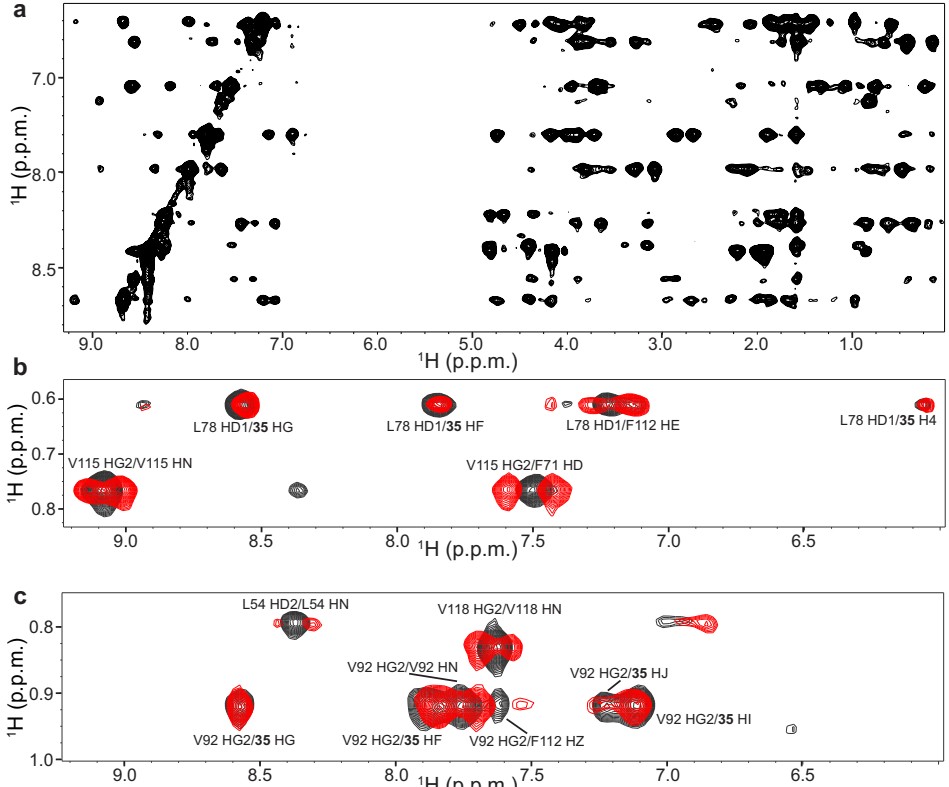

**Fig. 3 [1]H-[15]N and [1]H-[13]C NOESY-HSQC spectra of the Bcl-2-xL/35 complex yield high quality data and intermolecular NOEs. a** Two-dimensional [1]H-[1]H slice of a simultaneous 3D [1]H-[15]N, [1]H-[13]C NOESY-HSQC spectrum of the Bcl-2-xL/**35** complex taken at 121.5 ppm in the [15]N dimension. **b**, **c** Two-dimensional [1]H-[1]H slices of simultaneous 3D [1]H-[15]N, [1]H-[13]C NOESY-HSQC spectra of the Bcl-2-xL/**35** complex acquired with (black contours) or without (red contours) [15]N and [13]C decoupling during t1. The slices were taken at 24.2 ppm (**b**) or 23.1 ppm (**c**) in the [13]C dimension.

concentrations on the 200-300 μM range and also by some tendency to degradation. These problems were alleviated to some extent when Bcl-2-xL was bound to **35**, allowing acquisition of NMR spectra at this concentration range for periods of 3-5 days at 20 °C. Using multiple uniformly [15]N- and [15]N,[13]C-labeled samples, we were able to acquire a full set of spectra for structure determination of the Bcl-2-xL/**35** complex (see "Methods"). Triple resonance spectra typically used for backbone and side chain assignment (e.g. HNCACB or HCCH-TOCSY) were of poor quality under these conditions. However, these spectra provided sufficient information to help assign the majority of backbone resonances of Bcl-2-xL bound to **35** (Fig. S3) with critical help of the nuclear Overhauser effect (NOE) data obtained from 3D [1]H-[15]N NOESY-HSQC (Fig. 3a) and simultaneous 3D [1]H-[15]N, [1]H-[13]C NOESY-HSQC spectra, which were of high quality (Fig. 3b, c, black contours). We were also able to assign almost all the aromatic and methyl-containing side chains, as well as many of the polar side chains. Comparison of simultaneous 3D [1]H-[15]N, [1]H-[13]C NOESY-HSQC spectra with or without [13]C and [15]N decoupling during t1 evolution (Fig. 3b, c, black and red contours, respectively) allowed us to distinguish intermolecular NOES of Bcl-2-xL with **35** from intramolecular NOEs within Bcl-2-xL. The intermolecular NOEs, together with intramolecular correlations within **35** observed in [1]H-[1]H 2D NOESY and 2D TOCSY spectra that were [13]C,[15]N-filtered in both dimensions, yielded assignments of all the protons from **35** except for the carbamate NH group (H16; see hydrogen nomenclature used for structure determination in Fig. 1).

Structures of the Bcl-2-xL/**35** complex were calculated by simulated annealing with torsion angle dynamics using CNS[22].

Since the experiments that demonstrated the inhibition of Bcl-2/ Beclin 1 BH3 domain binding by **35** were performed with racemic **35**, it was unclear which of the two enantiomers constitutes the active compound. Using PRODRG[23], we built initial models of both enantiomers and obtained force field parameters for structure calculations with CNS. The scaling factor incorporated into the parameters yielded by PRODRG was adjusted to properly enforce the covalent geometry of **35** without the system breaking apart during the simulations, and separate calculations performed with either the R or the S enantiomer, together with analysis of intermolecular NOEs, showed that the S enantiomer of **35** is the bona fide ligand of Bcl-2-xL (see "Methods").

Final structures of the Bcl-2-xL/**35** complex were obtained using a total of 3124 experimental restraints, including 60 intermolecular NOEs (Tables 1, 2). The structure of the protein contains the eight α-helices that have been observed in previously determined structures of Bcl-2 (e.g.[12]) and is generally well defined except for flexible regions, which included the N-terminal tail, the long loop with the Bcl-xL sequence and a short loop following residue 165 that connects helix α8 with the C-terminal juxtamembrane region (Fig. 4a, b) (note that residue 165 corresponds to residue 206 of native Bcl-2 because the long flexible loop was replaced with a shorter sequence). The average root mean square (r.m.s.) deviation among the backbone atoms of the 20 structures, excluding the N-terminal tail and the flexible loop, is 0.56 Å (Table 2). The quality of the structure is also shown by low deviations from the experimental restraints and idealized covalent geometry, as well as by good Ramachandran plot statistics (Table 2; the 1.8% of residues in nonallowed regions are from the flexible regions where there are not sufficient NOEs to define the structure). The structure of **35**

**Table 1 Intermolecular distance restraints used in the structure elucidation of the Bcl-2-xl/35 complex.**

| Residue | hydrogen | 35 hydrogen | upper restraint (Å) |
|---------|----------|-------------|---------------------|
| F71 | HE* | HM* | 4.5 |
| F71 | HZ | HM* | 3.3 |
| F71 | HZ | HN* | 6 |
| M74 | HE* | HK* | 6.5 |
| M74 | HE* | HL* | 4.5 |
| M74 | HE* | HM* | 6 |
| L78 | HD1* | H4 | 5.5 |
| L78 | HD1* | HF* | 4.5 |
| L78 | HD1* | HG* | 4.5 |
| L78 | HD2* | H4 | 6.5 |
| L78 | HD2* | HF* | 6.5 |
| L78 | HD2* | HG* | 6.5 |
| L78 | HD2* | HH* | 6 |
| L80 | HD1* | HF* | 3.8 |
| L80 | HD1* | HG* | 4.5 |
| L80 | HD2* | HF* | 6.5 |
| A85 | HB* | HF* | 4.5 |
| A85 | HB* | HG* | 6.5 |
| R88 | HA | HJ* | 6 |
| R88 | HB* | HF* | 4.5 |
| R88 | HB* | HG* | 4.5 |
| R88 | HB* | HI* | 4.5 |
| T91 | HB | HJ* | 4 |
| T91 | HG2* | H9 | 5.5 |
| T91 | HG2* | HJ* | 3.8 |
| V92 | HG1* | HK* | 4.5 |
| V92 | HG1* | HL* | 3.8 |
| V92 | HG1* | HM* | 6 |
| V92 | HG1* | HN* | 3.8 |
| V92 | HG2* | HF* | 4.5 |
| V92 | HG2* | HG* | 3.8 |
| V92 | HG2* | HI* | 3.8 |
| V92 | HG2* | HJ* | 6.5 |
| L96 | HD1* | H23 | 5.5 |
| A108 | HB2 | H23 | 5.5 |
| A108 | HB* | HM* | 6.5 |
| F112 | HB* | HM* | 6.5 |
| F112 | HN | HM* | 4 |
| F112 | HN | HN* | 4 |
| F112 | HZ | HF* | 3.3 |
| F112 | HZ | HG* | 4 |
| M116 | HE* | HF* | 6.5 |
| M116 | HE* | HG* | 6.5 |
| F171 | HZ | H23 | 3.5 |
| F171 | HZ | HM* | 5.5 |
| F171 | HZ | HN* | 4 |
| W173 | HE1 | HK* | 7 |
| W173 | HE1 | HL* | 4 |
| L174 | HD1* | H23 | 5.5 |
| L176 | HB* | HH* | 5 |
| L176 | HD1* | HH* | 6 |
| L176 | HD1* | HI* | 4.5 |
| L176 | HD1* | HJ* | 6.5 |
| L176 | HD1* | HK* | 6.5 |
| L176 | HD2* | HH* | 4.5 |
| L176 | HD2* | HI* | 4.5 |
| L176 | HD2* | HJ* | 4.5 |
| L176 | HD2* | HK* | 6.5 |
| L176 | HN | HK* | 6 |
| L176 | HN | HL* | 4 |

**Table 2 NMR and structural statistics for the 20 simulated annealing structures of the Bcl-2-xl/35 complex with the lowest NOE energies.**

| NMR distance and dihedral restraints (3124 total)[a] | |
|---|---|
| Distance constraints | |
| Total NOE | 2738 |
| Intra-residue | 545 |
| Inter-residue | 2133 |
| Sequential ($|i - j| = 1$) | 655 |
| Medium-range ($|i - j| = 2$–4) | 714 |
| Long-range ($|i - j| > 4$) | 764 |
| Intermolecular | 60 |
| Hydrogen bonds | 116 |
| Total dihedral angle restraints | 270 |
| $\phi$ | 135 |
| $\psi$ | 135 |
| Structure statistics | |
| Violations (mean and s.d.) | |
| Distance constraints (Å) | $0.006 \pm 0.0002$ |
| Dihedral angle constraints (º) | $0.047 \pm 0.012$ |
| Max. dihedral angle violation (º) | <2º |
| Max. distance constraint violation (Å) | <0.2 Å |
| Deviations from idealized geometry | |
| Bond lengths (Å) | $0.0011 \pm 0.00004$ |
| Bond angles (º) | $0.38 \pm 0.005$ |
| Impropers (º) | $0.57 \pm 0.045$ |
| Average pairwise r.m.s. deviation (Å) | |
| Backbone residues (10–31, 47–176) | 0.56 |
| Heavy-atom residues (10–31, 47–176) | 1.29 |
| Backbone secondary structure[b] | 0.49 |
| Heavy-atom secondary structure[b] | 1.23 |

[a]There were no violations above 0.2 Å for NOE or hydrogen bond restraints, or above 2º for dihedral angle restraints.
[b]Seven α-helices, residues 10–26, 47–66, 68–77, 85–97, 103–123, 126–151, 154–160.

The structures of the top five models yielded by AlphaFold2 contained the expected eight α-helices and were basically identical to each other except at the C-terminal juxtamembrane region (residue 166-177), which adopts a different conformation in each model (Fig. 4c). Nevertheless, the juxtamembrane region folds back onto the globular domain in all models, having intramolecular interactions with helices α2, α3, α4 and α5. Hence, although AlphaFold2 did not yield a well-defined structure for the juxtamembrane region, the prediction that this region folds back onto the globular domain is consistent with the differences in the $^1$H-$^{15}$N TROSY-HSQC spectra caused by the presence of this region (Fig. 2a). Superposition of the top model of Bcl-2-xL generated by AlphaFold2 with the NMR structure of the Bcl-2-xL/35 complex with the lowest NOE energy shows that the two structures are very similar up to residue 165 (0.65 Å r.m.s. deviation between the backbone atoms of 139 structured residues), but the conformation of the juxtamembrane region is more extended in the Bcl-2-xL/35 complex and makes extensive contacts with 35 that are not possible for the isolated protein (Fig. 4d).

**Structural basis for recognition of 35 by Bcl-2 and selective inhibition.** Figure 5a, b illustrate how 35 binds to a highly hydrophobic area of Bcl-2 formed by helices α3, α4 and α5 together with the juxtamembrane region, which acts as a lid, burying most of 35 inside Bcl-2-xL (Fig. 5c). The bromophenyl group of 35 binds to a hydrophobic pocket formed by L78, L80, A85, R88, F112 and M116, while the more exposed phenyl group is surrounded by R88, T91, V92 and L176. The dihydropyrazol

bound to Bcl-2-xL is also well defined except for the orientation of the carbamate group (Fig. 4b).

Since we were unable to determine the structure of isolated Bcl-2-xL experimentally, we predicted its structure using AlphaFold2[24].

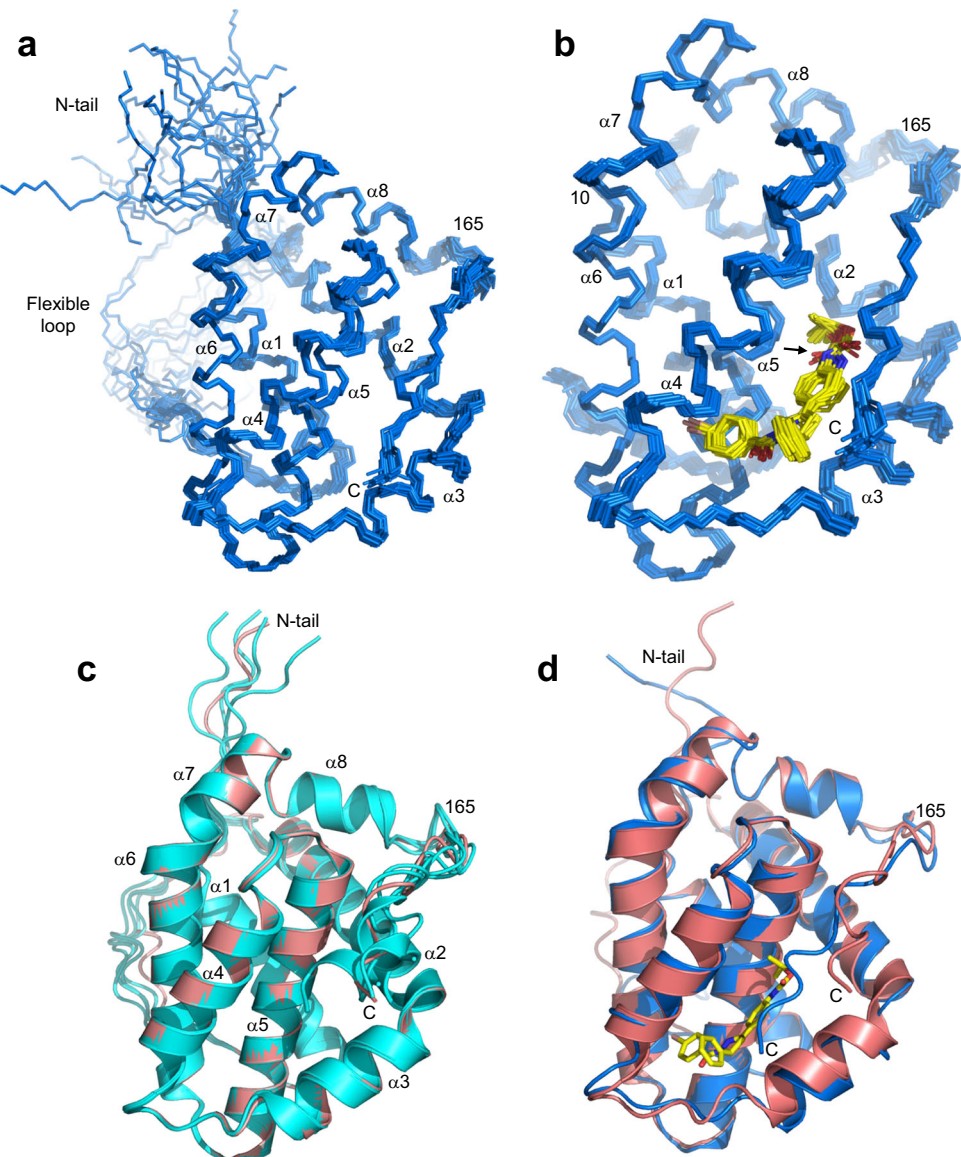

**Fig. 4 NMR structure of the Bcl-2-xL/35 complex. a** Superposition of the backbone of the 20 structures of the Bcl-2-xL/**35** complex with the lowest NOE energies. The positions of the flexible N-terminal tail (N-tail), the C-terminus (C), the eight α-helices (labeled α1-α8) and the flexible loop connecting α1 to α2 are indicated. **b** Backbone superposition of the structured parts of the Bcl-2-xL/**35** complex (blue) with the bound **35** molecule represented by stick models (carbon atoms in yellow, nitrogen atoms in blue, oxygen atoms in red and bromine atoms in pink). The arrow indicates the position of the carbamate group, which adopts different orientations in the structures. **c** Superposition of ribbon diagrams from the top five structures of Bcl-2-xL predicted by AlphaFold2[24]. The top model is colored in salmon. **d** Superposition of ribbon diagrams of the top AlphaFold2 model of Bcl-2-xL and the structure of the Bcl-2-xL/**35** complex with the lowest NOE energy. The position of residue 165, which corresponds to residue 206 of native Bcl-2, is indicated in all panels.

ring contacts L78, V92 and L176, whereas the central phenyl group is sandwiched between M74, L78, V92, F112, W173 and L176. Finally, the isopropyl group binds to a pocket formed by F71, M74, V92, A108, F112 and F171. The view of Fig. 5d, in which the surface of the juxtamembrane region has been removed, shows that the globular domain of Bcl-2-xL forms the two distal hydrophobic pockets where the bromophenyl and isopropyl groups of **35** bind, as well as a small hydrophobic groove where the central dihydropyrazole and phenyl rings of **35** bind. Hence, it seems likely that **35** can bind to Bcl-2 without the juxtamembrane region, but with lower affinity because the interactions of the juxtamembrane region with **35** contribute substantially to the overall affinity. The weaker binding of **35** to Bcl-2(1-206) may not be observable in our NMR experiments

(Fig. 2b) because the insolubility of **35** limits the concentrations of free **35** that can be reached.

The binding site of **35** on Bcl-2-xL is clearly different from that of ABT-737 (Fig. 6a). While the binding site of ABT-737 extends over the upper part of the BH3-binding groove in the orientation of Fig. 6a, **35** binds to the bottom part of this groove, and there is only a small degree of overlap between the two sites. Indeed, inspection of multiple structures of Bcl-2 or Bcl-xL bound to distinct inhibitors shows that none of them bound to the bottom of the groove (Fig. 6b). Interestingly, the comparison of the binding modes of the Beclin 1 and Bax BH3 domains to Bcl-2 shows that, although their binding modes are similar as expected because of their homology, the Beclin 1 BH3 domain has an additional helical turn at the N-terminus that binds to the bottom

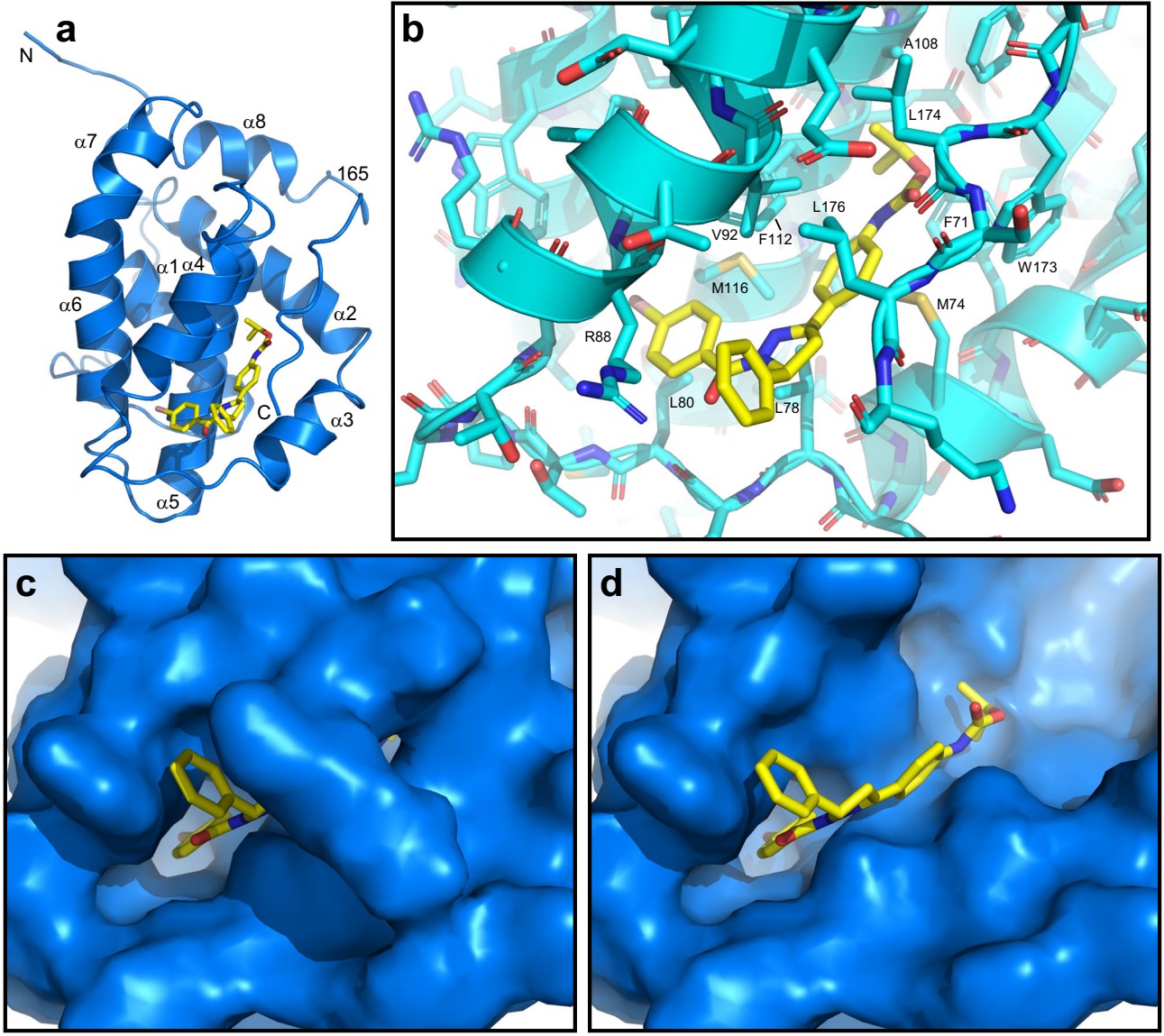

**Fig. 5 Binding mode of 35 to Bcl-2-xL. a** Ribbon diagram of the structure of the Bcl-2-xL/**35** complex with the lowest NOE energy with the Bcl-2-xL ribbon in blue and **35** represented by stick models (carbon atoms in yellow, nitrogen atoms in blue, oxygen atoms in red and bromine atoms in pink). The positions of the N- and C-termini are labeled N and C, respectively, and the eight α-helices are labeled α1-α8. The position of residue 165, which corresponds to residue 206 of native Bcl-2, is indicated. **b** Close-up view of the **35** binding site with Bcl-2-xL represented by a cyan ribbon and stick models (carbon atoms in cyan, nitrogen atoms in blue, oxygen atoms in red and sulfur atoms in light orange). Selected Bcl-2-xL side chains that contact **35** are labeled. **c, d** Close up view of the **35** binding site with the molecular surface of Bcl-2-xL shown in blue and **35** represented by stick models. In (**d**), the surface of the juxtamembrane region at the C-terminus of Bcl-2-xL was removed such that the binding site of **35** on the globular domain can be observed.

of the groove and is not present in the Bax BH3 domain (Fig. 6c, d). Contacts between Bcl-2-xL and this extra helical turn of Beclin 1 BH3 are mediated by backbone atoms, the Thr108 methyl group and the Met109 side chain of Beclin 1, which bind to a similar region as the most exposed phenyl group of **35** (Fig. 6d, e). Note also that more than half of the area of Bcl-2 that contacts **35** does not participate in binding to the Bax BH3 domain (Fig. 6f). These findings suggest that the much stronger inhibition of Beclin 1 BH3/Bcl-2 binding by **35**, compared to Bax BH3/Bcl-2 binding[20], arises in part because **35** targets a pocket of Bcl-2 that interacts selectively with Beclin 1 but not with Bax.

## Discussion
The roles of autophagy in a wide variety of cellular processes, together with the phenotypes caused by deletion of proteins that promote autophagy or by mutations that enhance autophagy,

have led to the belief that stimulating autophagy might provide powerful therapies against multiple diseases[1,2,4–6]. Because these include some of the most devasting human diseases such as cancer and neurodegeneration, it is urgent to identify drugs that increase autophagy effectively and selectively. Bcl-2 is an attractive target for this purpose because of its role in inhibiting autophagy by sequestering Beclin-1[2], and the potential health benefits of targeting this interaction were supported by the phenotypes observed in knockin mice bearing a mutation in Beclin 1 that disrupts binding to Bcl-2[7,8]. However, a key concern is whether it is possible to disrupt binding of Beclin 1 BH3 to Bcl-2 without triggering apoptosis by inhibiting interactions between Bcl-2 and BH3-containing proapoptotic proteins. Our recent discovery of compounds that inhibit Bcl-2/Beclin 1 BH3 binding much more potently than Bcl-2/Bax BH3 binding supported the feasibility of this approach. The structure of the Bcl-2-xL/**35**

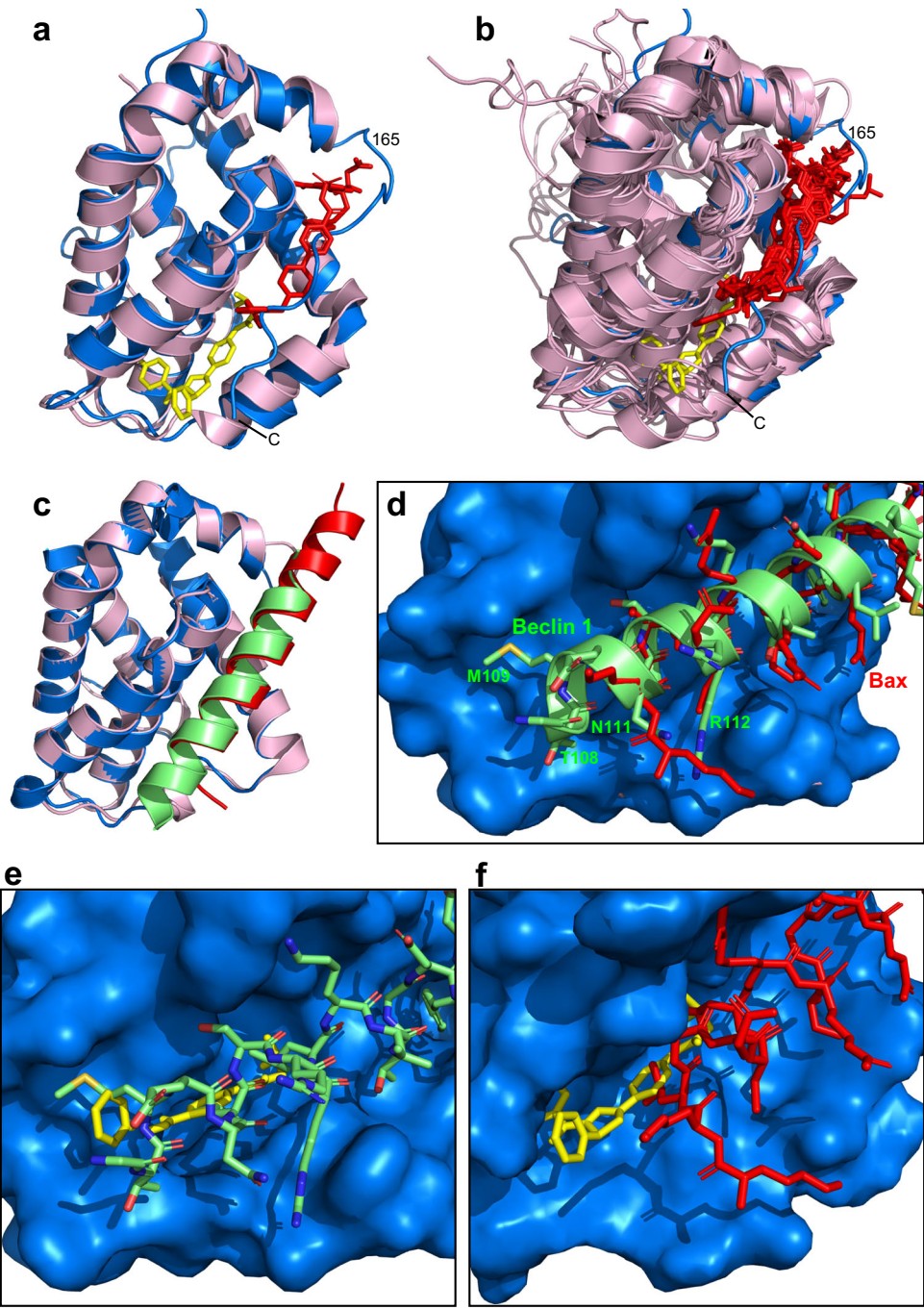

complex presented here now reveals that much of the **35** binding site is formed by a surface of Bcl-2 that does not participate in binding to other Bcl-2 inhibitors described previously and that is involved in binding to the Beclin-1 BH3 domain but not the Bax BH3 domain. These results further support the notion that it may be possible to develop Bcl-2-binding drugs for selective activation of autophagy in disease and provide a fundamental roadmap to achieve this goal.

Three key properties of **35** that are interrelated and that need to be optimized for compounds in this family to have a realistic chance of clinical applications are solubility, activity and specificity of Bcl-2/Beclin 1 inhibition. A key consideration for optimization of these properties is the role of the juxtamembrane region, as it is unclear whether this region can participate in binding to **35** or related compounds in a native environment with Bcl-2 anchored on a membrane. Thus, the juxtamembrane region

most likely interacts strongly with the membrane because of its highly hydrophobic nature and its proximity to the lipids, which would prevent interactions of the juxtamembrane region with an organic compound bound to the globular domain. Binding of the juxtamembrane region to the lipids might also hinder the intramolecular binding of the juxtamembrane region to the globular domain revealed by comparison of the $^{1}$H-$^{15}$N TROSY-HSQC spectra of Bcl-2(1-206) and Bcl-2(1-218) (Fig. 2a). We have attempted to address these questions by NMR analyses of full-length Bcl-2 incorporated into nanodiscs, but these attempts have been hindered so far by the high tendency of full-length Bcl-2 to precipitate in detergents commonly used for nanodisc preparation, perhaps because of the instability of the protein.

Even with the uncertainty about the participation of the juxtamembrane region in binding of membrane-anchored Bcl-2 to potential drugs, the Bcl-2-xL/**35** binding mode revealed by our

**Fig. 6 The 35 binding site in Bcl-2-xL is distinct from those of previously described Bcl-2 inhibitors and overlaps with the unique site that binds to the Beclin 1 BH3 domain N-terminus but not to Bax BH3. a** Superposition of ribbon diagrams of the Bcl-2-xL/**35** complex (blue ribbon with **35** represented by yellow sticks) and the complex of a chimera in which the long flexible loop of Bcl-2(1-207) was replaced by a shorter sequence from the homologous loop of Bcl-xL [below referred to as Bcl-2-xL(1-207)] with ABT-737 (pink ribbon with ABT-737 represented by red sticks) (PDB accession number 6QGG). The position of residue 165, which corresponds to residue 206 of native Bcl-2, is indicated. **b** Superposition of ribbon diagrams of the Bcl-2-xL/**35** complex (blue ribbon with **35** represented by yellow sticks) and various structures of Bcl-2-xL(1-207) or Bcl-xL bound to distinct inhibitors (pink ribbon with the inhibitors represented by red sticks) (PDB accession numbers 1YSG, 1YSN, 1YSW, 3INQ, 3QKD, 4IEH, 4LVT, 4LXD, 4MAN, 4QNQ, 6QGH, 6QGG, 7JMT). **c** Superposition of ribbon diagrams of Bcl-xL (blue ribbon) bound to the Beclin 1 BH3 domain (lime ribbon) with the structure of Bcl-2-xL(1-207) (pink ribbon) bound to the Bax BH3 domain (red ribbon) (PDB accession numbers 2P1L and 2XA0, respectively). Note that the structure of Bcl-2-xL(1-207) bound to the Beclin 1 BH3 domain (PDB accession number 5VAU) is very similar to that of the Bcl-xL/Beclin 1 BH3 complex; we show the latter because the former lacks T108 at the N-terminus of the BH3 domain. **d** Close-up view of the binding site of the N-termini of the BH3 domains (same structures as in **c**), with the molecular surface of Bcl-xL shown in blue, the Bax BH3 domain represented by red sticks and the Beclin 1 BH3 domain represented by a lime ribbon and sticks (carbon atoms in lime, nitrogen atoms in blue, oxygen atoms in red and sulfur atoms in yellow). Bcl-2-xL(1-207) is not shown. **e** Close-up view of the binding site of the Beclin 1 BH3 domain N-terminus (represented by sticks) in the molecular surface of Bcl-xL (same coloring as **d**) superimposed with the structure of the Bcl-2-xL/**35** complex (**35** represented by yellow sticks; Bcl-2-xL not shown). **f** Close-up view of the binding site of **35** (yellow sticks) on the molecular surface of the globular domain of Bcl-2-xL (blue; juxtamembrane region removed) superimposed with the structure of the Bcl-2-xL(1-207)/Bax BH3 complex (Bax BH3 represented by red sticks; Bcl-2-xL(1-207) not shown). Panels (**e**, **f**) show that much of the **35** binding site overlaps with the binding site of the Beclin 1 BH3 N-terminus that does not interact with the Bax BH3 domain.

structure provides key clues on how to develop such drugs, for instance by providing guidance on how to improve the solubility. Thus, a limitation of our study is that we did not assess whether **35** can selectively activate autophagy in cells, which is hindered by the high insolubility of this compound in aqueous buffers at physiological pH. Indeed, no binding of **35** to Bcl-2(1-218) was detected using $^1$H-$^{15}$N TROSY-HSQC spectra upon addition of stoichiometric amounts at 50–70 μM protein concentration. Preparation of NMR samples with Bcl-2(1-218) or Bcl-2-xL bound to **35** required addition of a large excess of **35** (5 to 10 fold) to 1 μM protein followed by concentration, and in some cases cross-peaks corresponding to free protein still remained. In such cases, the procedure was repeated to reach saturation. The structure of the Bcl-2-xL/**35** complex shows that a natural way to improve solubility is to attach a polar group (perhaps charged) to the phenyl ring of **35** that 'snorkels' out of the Bcl-2-xL cavity (Fig. 5c).

If the juxtamembrane region can participate in binding to organic compounds in vivo, the affinity of **35** or similar compounds for Bcl-2 is likely sufficient to stimulate autophagy in vivo as long as they are soluble enough. Conversely, it is crucial to develop **35** analogues with enhanced affinity for Bcl-2 if the juxtamembrane region cannot participate in binding in a native environment. The binding mode between **35** and the Bcl-2 globular domain revealed by our structure (Fig. 5d) suggests avenues to improve the affinity of this family of compounds for Bcl-2 in the absence of energetic contributions from the juxtamembrane region to binding. For instance, attachment of chemical groups (perhaps including aromatic rings) to the dihydropyrazole ring and/or the more exposed phenyl group at one end of **35** could provide stabilizing interactions with nearby surfaces at the very bottom of the BH3-binding groove, some of which are involved in binding to the N-terminal residues of the Beclin 1 BH3 domain (Fig. 6e). The added moieties could bear polar groups to improve solubility. Affinity and/or solubility could also be improved by attaching an additional chemical moiety to the exposed side of the central phenyl group of **35** linked to the carbamate group. Note also that the bromphenyl group of **35** is tucked into a cavity that is not normally observed or is smaller in structures of Bcl-2 bound to other compounds or BH3 domains. The formation or enlargement of this cavity upon **35** binding likely reflects malleability in this region of Bcl-2. Such malleability may also allow the formation of other cavities if an appropriate chemical group is attached to the **35** framework at the appropriate location. Importantly, tests of new compounds

designed with these ideas should be explored with a Bcl-2 fragment lacking the juxtamembrane region such that this region does not interfere with studies of structure-activity relationships.

In principle, the high selectivity of **35** in inhibiting Bcl-2/Beclin 1 BH3 binding compared to Bax BH3 binding (200-fold)[20] might arise merely because the Bax BH3 domain binds to Bcl-2 with much higher affinity ($k_D$ 15 nM[15]) than the Beclin 1 BH3 domain ($k_D$ 1–3 μM[14,16]). However, the selectivity of ABT-737 in inhibiting Bcl-2-Beclin 1 binding versus Bcl-2-Bax binding is only 20-fold[20], and more than half of the **35** binding site involves an area of Bcl-2 that contributes to Beclin 1 binding but not Bax binding (Fig. 6e, f). These observations suggest that the selectivity of **35** as an inhibitor of Bcl-2—Beclin 1 binding arises in part because **35** targets this area of Bcl-2. The proposed addition of chemical groups to the dihydropyrazole ring and/or the more exposed phenyl group at one end of **35** to expand the interactions in this area and enhance affinity would be expected to further enhance the selectivity for disruption of Bcl-2/Beclin 1 interactions compared Bcl-2/Bax interactions. With enhanced affinity at this end, it might even be possible to increase selectivity by dispensing with part of **35** at the other end that invades the Bax binding site (e.g. the isopropyl group; Fig. 6f). It is also worth noting that the Thr108-Met109 sequence of Beclin 1 is generally not present in other BH3 domains (Fig. S4). Although the human Bak BH3 domain contains a reversed Thr-Met dyad at this position, these residues were not necessary for binding of Bak BH3 to Bcl-xL[13] and are not conserved in mouse Bak (Fig. S4), which binds tightly to Bcl-2 ($k_D$ 70 nM)[15]. Nevertheless, it is important to note that the helices of the BH3 domains from some pro-apoptotic proteins such as Bid or Bim bound to Bcl-2 or Bcl-xL are extended at the N-terminus to similar degrees as the Beclin 1 BH3 helix (Fig. S5). In addition, activation of autophagy without inducing apoptosis requires selective inhibition of Beclin 1 binding to not only Bcl-2 but also to other Bcl-2 homologues that control apoptosis such as Bcl-xL and Mcl-1. Hence, the notion that **35** and/or related compounds might act as selective stimulators of autophagy needs to be further tested with systematic studies of how these compounds compete with binding of a variety of pro-apoptotic proteins and Beclin 1 to the relevant members of the Bcl-2 family, which depend on the cell type. A first important step to perform such studies is to obtain **35** analogues with improved solubility, which will facilitate the performance of quantitative binding and competitions assays.

In retrospective, it is paradoxical that the presence of the juxtamembrane region helped to discover selective inhibitors of

Bcl-2/Beclin 1 binding[20] but this region is likely unnecessary for further development of compounds that might stimulate autophagy by targeting this interaction. The structural framework provided by the NMR structure of Bcl-2-xL/**35** described here paves the way for progress in this area.

## Methods

**Protein expression and purification**. A pSKB2 vector to express residues 1-218 of human Bcl-2 (isoform α) with a N-terminal His₆-tag and an HRV 3 C cleavage site was described previously[20]. An analogous vector to express residues 1-206 of Bcl-2 was a kind gift from Beth Levine (UT Southwestern). A pGEX4T1 vector to express the Bcl-2-xL chimera, in which the long flexible loop of Bcl-2(1-218) (residues 35-91) was replaced by a shorter sequence from the homologous loop of Bcl-xL (residues 29−44) was prepared using standard recombinant DNA techniques with custom designed primers. The vector expresses Bcl-2-xL fused at the N-terminus to glutathione S-transferase (GST) with a TEV (Tobacco Etch Virus protease) cleavage site to remove the GST moiety.

To prepare ¹⁵N-labeled His₆-tagged Bcl-2 (1-206 or 1-218), pSKB2-Bcl-2 (1-206 or 1-218) was used to transform BL21 (DE3) phage resistant *E. coli* competent cells (Invitrogen). A selected expression clone was cultured in M9 minimal medium containing ¹⁵N-labeled ammonium chloride (Cambridge Isotope Laboratories) as the sole nitrogen source. The cells were grown at 37 °C to 0.8 optical density at 600 nm (OD600), and then induced by addition of 0.4 mM isopropyl-β-D-thiogalactopyranoside (IPTG) followed by overnight culture at 25 °C. Bacteria were collected by centrifugation and the pellet was resuspended in a buffer containing 50 mM Tris (pH 8.0), 150 mM NaCl, 20 mM imidazole and 1 mM TCEP [tris(2-carboxyethyl)phosphine], and protease inhibitors (Sigma-Aldrich, P2714-BTL). The cells were then passed four times through an Emulsiflex-C5 (AVESTIN) cell disruptor with pressure and centrifuged at 20,000 rpm for 30 min at 4 °C to remove cell debris. The protein in the supernatant was purified using Ni-NTA column (GE Healthcare) by equilibrating with 20 mM Tris (pH 8.0), 150 mM NaCl, 1 mM TCEP (buffer A) containing 20 mM imidazole, washing the unbound impurities by adding 1 M NaCl in buffer A and buffer A containing 1% triton X-100, equilibrating again with buffer A containing 20 mM imidazole, and eluting the protein with 250 mM imidazole, 50 mM Tris (pH 8.0), 150 mM NaCl, and 1 mM TCEP. The sample was placed into a 3.5 kDa molecular weight cutoff (MWCO) dialysis bag (Fisher brand) and 3 C precision protease was added to cleave the N-terminal His₆-tag. The sample was dialyzed against 2 L of buffer A overnight at 4 °C and passed again through a Ni-NTA column to remove 3 C precision protease or uncut protein. The sample was further purified by size exclusion chromatography in 20 mM Tris, pH7.4, 150 mM NaCl, 1 mM TCEP using a Superdex 75 10/600 column (GE Healthcare). The purified protein was concentrated by centrifugation using an Amicon Ultra-15 Centrifugal Filter Unit with a 3.5 KDa MWCO (EMD Millipore).

The procedures used to express uniformly ¹⁵N- or ¹⁵N,¹³C-labeled Bcl-2-xL in bacteria and purify the protein were analogous except that ¹³C-labeled glucose was used as the sole carbon source for ¹⁵N,¹³C-labeling and that before gel filtration the protein was purified by affinity chromatography on Glutathione Sepharose 4B (GE Healthcare), equilibrating the cell supernatant on the beads with buffer A, washing the beads with buffer A containing 1 M NaCl, buffer A containing 1% triton X-100 and buffer A, and cleaving the GST moiety on the resin with TEV at 4 °C for overnight on rotator.

All purified proteins were stored in small aliquots at −80 °C after flash freezing in liquid nitrogen. Protein concentrations were determined from the absorption at 280 nm and the purity of protein was examined by SDS-PAGE and Coomassie blue staining. L-amino acid peptides were synthesized by the Protein Chemistry Technology Core at the University of Texas Southwestern Medical Center. The Beclin 1 BH3 peptide includes residues 105-130 of human Beclin 1 protein: DGGTMENLSRRLKVTGDLFDIMSGQT. Bax BH3 peptide includes residues 49-84 of human Bax: PVPQDASTKKLSECLK-RIGDELDSNMELQRMIAAVD. All peptides were purified by HPLC with an S4 column (purity over 95%) and the identity was validated by mass spectrometry. Peptides were dissolved in d₆-dimethylsulfoxide and stock solutions were stored in small aliquots at −80 °C.

**Crystallization trials**. A Gryphon Crystal Screenings robot was used to perform all crystallization trials. The temperature for screening was set at 20 ºC and a PCT crystallization screen (Hampton Research) was performed for each protein or complex to determine the optimal protein concentration for crystallization trials. The following screen kits (Hampton Research) with the indicated protein or complexes and concentrations were used in the trials: (i) Index, JCSG, Crystal, PACT, Clear Strategy, Wizard, PEG/Ion, PEG/RxHT and HT-96 ProPLex for Bcl-2(1-218) alone (5 mg/ml); (ii) Index, JCSG, Crystal, PACT and Clear Strategy for Bcl-2 (1-218) alone (9.4 mg/ml); (iii) JCSG and Crystal screen HT for Bcl-2(1-218)/**35** complex (12 mg/ml); (iv) Index, JCSG, Crystal, PACT, Clear Strategy, PEG/Ion, and PEG/RxHT for Bcl-2-xL alone (7 mg/ml); and (v) Index, JCSG, Crystal screen HT, PACT, Clear Strategy and PEG/Ion, and PEG/RxHT for Bcl-2-xL/**35** complex (13.5 mg/ml). The complexes were prepared as described below for NMR samples and changes on the methyl region caused by **35** binding in 1D ¹H NMR spectra were used to assess quantitative binding of the compound to the protein.

**NMR spectroscopy**. NMR spectra were acquired on Agilent DD2 spectrometers equipped with cold probes and operating at 600 MHz (triple resonance spectra for resonance assignments) or 800 MHz (all other spectra). ¹H-¹⁵N TROSY-HSQC and HSQC spectra were acquired at 25 °C using solutions of 10–45 µM uniformly ¹⁵N-labeled protein in 20 mM Sodium Phosphate (pH 7.3) containing 1 mM TCEP and 7% D₂O. ABT-737 or Beclin 1 BH3 domain were added directly to the protein sample from concentrated stocks in d₆-DMSO. Because of the insolubility of **35**, to prepare NMR samples containing this compound the protein was diluted to 1 µM and **35** (dissolved in d₆-dimethylsulfoxide or deuterated acetonitrile/deuterated dimethylsulfoxide 3.5: v/v) was added slowly and with stirring to a final concentration of 5–10 µM. The samples were concentrated to 10–45 µM protein concentration before acquiring the spectra.

Uniformly ¹⁵N- or ¹⁵N,¹³C-labeled samples of Bcl-2-xL/**35** complex for structure determination were prepared by an analogous procedure but concentrating the complex to 200–300 µM. The samples exhibited a tendency to aggregation and degradation, but they were sufficiently stable to acquire NMR data for 3–5 days at 20 °C. Resonance assignments were obtained from sensitivity enhanced 2D ¹H-¹⁵N HSQC, 3D HNCO, 3D HNCACB, 3D CBCACONH, 3D (H)C(CO)NH-TOCSY and 3D H(C)(CO)NH-TOCSY, together with standard homonuclear 2D NOESY and 2D TOCSY, ¹⁵N,¹³C-¹⁵N,¹³C-filtered homonuclear 2D TOCSY and 2D NOESY, 2D ¹H-¹³C HSQC, 3D HCCH-TOCSY, simultaneous 3D ¹H-¹⁵N, ¹H-¹³C NOESY-HSQC, 2D (HB)CB(CGCD)HD and 2D (HB)CB(CGCDCE)HE spectra[25–29]. Stereospecific assignments of Val and Leu methyl groups were

obtained from a constant-time $^1$H-$^{13}$C HSQC spectra acquired on a sample in which Bcl-2-xL was 10% $^{13}$C-labeled[30]. All spectra were acquired in 20 mM Sodium Phosphate (pH 7.3) containing 1 mM TCEP and 7% D$_2$O, except for the standard homonuclear 2D NOESY and 2D TOCSY, which were acquired in the same buffer but with 100% D$_2$O. A $^1$H-$^{15}$N HSQC spectra of a freshly prepared sample dissolved in the same buffer in D$_2$O was used to determine which NH groups were protected from exchange with the solvent. The resonances of **35** were assigned from the 2D NOESY and 2D TOCSY spectra, the $^{15}$N,$^{13}$C-$^{15}$N,$^{13}$C-filtered homonuclear 2D TOCSY and 2D NOESY, and comparison of simultaneous 3D $^1$H-$^{15}$N, $^1$H-$^{13}$C NOESY-HSQC spectra acquired with or without $^{15}$N,$^{13}$C decoupling during t1, which yielded intermolecular NOEs between **35** and Bcl-2xL. All NMR data were processed with NMRpipe[31] and analyzed with NMRView[32].

**Structure determination**. Structures of the Bcl-2-xL/**35** complex compatible with experimental restraints were calculated using torsion angle simulated annealing with CNS[22] version 1.2. NOE cross-peak intensities were classified as strong, medium, weak and very weak, and assigned to interproton distance restraints of 1.8–2.8, 1.8–3.5, 1.8–5.0 and 1.8–6.0, respectively, with appropriate pseudoatom corrections using r$^{-6}$ averaging. A different scale was used for NOEs involving methyl groups to account for the different number of hydrogens involved. Phi and psi torsion angle restraints were derived from analysis of HN, $^{15}$N, $^{13}$Cα, $^{13}$CO, $^{13}$Cβ chemical shifts using TALOS[33], setting the restraints to the angle yielded with TALOS plus/minus the maximum of 20º or 1.5 times the standard deviation given by TALOS. Structures of Bcl-2-xL were initially calculated without **35** using distance restraints based on unambiguously assigned NOEs and torsion angle restraints. Hydrogen bond restraints were incorporated later based on the H/D exchange data and examination of preliminary structures, setting the H/O distance restraints to 1.7–2.5 Å and the N/O distance restraints to 2.7–3.5 Å. To incorporate **35** into the calculations, we used PRODRG[23] to build models of both enantiomers and obtained force field parameters, which are provide by PRODRG with a scaling factor to adjust the strength of the force constants. Torsion angle dynamics calculations of the Bcl-2-xL/**35** complex stopped abruptly when the scaling factor was set to 1, likely because high energies in the system, whereas the integrity of the **35** molecule broke apart when setting the scaling factor too low. Setting the scaling factor to 0.2 represented a compromise that allowed structure calculations.

We performed separate structure calculations incorporating either the R or the S enantiomer of **35** and 24 unambiguous intermolecular NOEs between the protein and the ligand. The ten lowest NOE energies and overall energies of the best structures obtained in the calculations performed with the S enantiomer (19–22 and 172–194 kcal/mol, respectively) were considerably lower than those obtained in calculations with the R enantiomer (22–26 and 200–213 kcal/mol). Interestingly, the structure with the lowest NOE energy obtained in calculations with the R enantiomer actually had a high energy (242 kcal/mol) because the ligand underwent an inversion to the S configuration during the calculation. Consistent with these observations, visual inspection of the best structures showed that the intermolecular NOEs around the chiral center could be accommodated much more readily with the S enantiomer without straining the covalent geometry of the ligand. Hence, we concluded that the S enantiomer of **35** is the bona fide ligand of Bcl-2-xL. Final structure calculations of the Bcl-2-xL/**35** complex were performed with 2738 NOE restraints, which included 60 intermolecular

NOE restraints, 270 dihedral angle restraints and 116 hydrogen bond restraints (Table 2). A total of 4497 structures were calculated and the 20 structures with the lowest NOE energies (ranging from 5.8 to 7.6 kcal/mol) were selected. The structural statistics are summarized in Table 2. Ramachandran plot statistics calculated with Molprobity: residues in favored regions 88.7%, residues in additionally allowed regions 9.5%, residues in non-allowed regions 1.8%. All residues in nonallowed regions are located in the flexible N-terminal tail, in the long flexible loop formed by residues 34-46, or in the loop formed by residues 162-267, for which there were insufficient assigned NOEs to accurately define its structure. The PyMOL Molecular Graphics System (Schrödinger, LLC) was used for visualization of the structures and preparation of related figures. The coordinates of the 20 structures have been deposited in the Protein Data Bank (accession ID 8U27) and the resonance assignments have been deposited in the BioMagResBank (accession ID 52100). Please note that the validation report prepared by the Protein Data Bank lists multiple violations of distance restraints above 0.2 Å because r$^{-6}$ sum is used to prepare the report instead of the r$^{-6}$ averaging used to calculate our structures. Hence, there are no real violations above 0.2 Å in the reported structures, as indicated in Table 2.

**Reporting summary**. Further information on research design is available in the Nature Portfolio Reporting Summary linked to this article.

## Data availability

The coordinates of the 20 structures of the Bcl-2-xL/**35** complex together with the restraints used for structure calculations are available at the Protein Data Bank (accession ID 8U27). The resonance are available at the BioMagResBank (accession ID 52100). Other data associated with this study are available from the corresponding author upon reasonable request.

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

## Acknowledgements

This paper is dedicated to the memory of Beth Levine, whose conviction that it is possible to selectively inhibit Beclin 1/Bcl-2 binding motivated the work presented here. We thank Xiaonan Dong for fruitful discussions in the beginning of this project, Liwei Wang, Yi-Chun Kuo and Xuewu Zhang for providing protein samples for initial NMR experiments, Lewis Kay for providing pulse sequences for NMR experiments and Axel Brunger for advice on how to include **35** in structure calculations with CNS. The Agilent DD2 console of the 800 MHz spectrometer used for the research presented here was purchased with a shared instrumentation grant from the NIH (S10OD018027 to J.R.). J.K.D.B. holds the Julie and Louis Beecherl, Jr., Chair in Medical Science. This work was supported by CETR grant U19AI142784 from the NIH (to Skip Virgin), sponsored research agreement SRA202201-0002 with VIR Biotechnology Inc and Welch Foundation grants I-1304 (to J.R.) and I-1422 (to J.K.D.B.).

## Author contributions

Y.-Z.P. and J.R. conceived the study. Y.-Z.P., Q.L., D.R.T., J.K.deB. and J.R. designed experiments. Y.-Z.P. prepared proteins and samples for NMR analysis and structure determination. Q.L. prepared compound **35**. Y.-Z.P. and J.R. elucidated the NMR structure of the Bcl-2-xL/**35** complex. J.R. wrote the manuscript. All authors proof-read and approved the manuscript.

## Competing interests

The authors declare no competing interests.
