## [Peer Review File · Communications Biology]

REVIEWERS' COMMENTS:

Reviewer #1 (Remarks to the Author):

The authors present have improved their MS upon reading the comments by 3 reviewers. The MS is now better and potentially suitable for publication in Communication Biology. Nevertheless, I still wonder if the authors responded carefully to all comments or if the main response was mainly in the reply letter and not so much on the MS side with no additional experiments or at least arguments why not. e.g.

A) s. response to review 3 in reply letter under 2) and 3).

Review: For example, the authors should address whether compound 35 can displace interaction of BCL-2/Beclin 1 complex and to less extend the BCL-2/BAX complex in cells. If that's the case, then they can demonstrate whether compound 35 can induce autophagy and not apoptosis.

Authors reply: As we mention in the manuscript and in the general comments above, compound 35 exhibits important solubility problems that hinder its use in meaningful cellular studies, and one of the important aspects of the structure described in our paper is the insights offered to improve compound solubility (lines 274-282)

My comment: Since most drugs are extremely hydrophobic like e.g. Venetoclax which is used as drug to inhibit Bcl-2 at its BH3 binding site (where also Beclin-1 can bind), I wonder how all the others were carrying out their cell assays with those drugs. since they have done it there must be a way for it. Nevertheless it might be outside the direct scope of the paper. But as the referees mentioned a couple of times: without cell data for this compound a lot of impact is lost.

B) similar mechanism of discussing away with the point with the membrane environment as done in reply to Reviewer 1 points 2 and 3:

my comments: Only because many studies were done in solution (with often truncated chimeric protein to induce solubility into membrane proteins) it does not mean one should ignore the fact that those proteins are anchored or even embedded in the mitochondrial outer membrane. And running solution NMR on their protein in membrane mimicking micelles might easily give them some ideas if their solution view is reflected in a more native-like environment. And titration with compound 35 one could try using small amounts of DMS e.g. like many people do.

Reviewer #2 (Remarks to the Author):

The authors have responded to my concerns with additional discussion and some additional in silico analyses.

I have only one comment. The idea that 35 is a proof of concept that the Bcl-2-Beclin1 interaction can be disrupted to induce autophagy without inducing apoptosis is no longer supportable, given that other BH3's bind to Bcl-2 similarly to Beclin, and no additional data are provided regarding the effect of 35 on such binding. I suggest that this may still be a possibility, but the data do not strongly support the idea either way.

Responses to the reviewer comments

REVIEWERS' COMMENTS:

Reviewer #1 (Remarks to the Author):

The authors present have improved their MS upon reading the comments by 3 reviewers. The MS is now better and potentially suitable for publication in Communication Biology. Nevertheless, I still wonder if the authors responded carefully to all comments or if the main response was mainly in the reply letter and not so much on the MS side with no additional experiments or at least arguments why not. e.g.

A) s. response to review 3 in reply letter under 2) and 3).

Review: For example, the authors should address whether compound 35 can displace interaction of BCL-2/Beclin 1 complex and to less extend the BCL-2/BAX complex in cells. If that's the case, then they can demonstrate whether compound 35 can induce autophagy and not apoptosis.

Authors reply: As we mention in the manuscript and in the general comments above, compound 35 exhibits important solubility problems that hinder its use in meaningful cellular studies, and one of the important aspects of the structure described in our paper is the insights offered to improve compound solubility (lines 274-282)

My comment: Since most drugs are extremely hydrophobic like e.g. Venetoclax which is used as drug to inhibit Bcl-2 at its BH3 binding site (where also Beclin-1 can bind), I wonder how all the others were carrying out their cell assays with those drugs. since they have done it there must be a way for it. Nevertheless it might be outside the direct scope of the paper. But as the referees mentioned a couple of times: without cell data for this compound a lot of impact is lost.

Communications Biology expressed an interest in publishing our paper without the need to perform additional experiments, recognizing the importance of the results that we describe by themselves. We do not have experience with Venetoclax, but the related compound ABT-737, which we used in our work, is much more soluble than compound **35**. We did point out in the manuscript this solubility problem, which is still mentioned in the revised manuscript along with a new statement pointing out that the lack of cellular data is a limitation of our work (in page 13, in the middle of the third paragraph of the discussion):

'Thus, a limitation of our study is that we did not assess whether **35** can selectively activate autophagy in cells, which is hindered by the high insolubility of this compound in aqueous buffers at physiological pH.'

B) similar mechanism of discussing away with the point with the membrane environment as done in reply to Reviewer 1 points 2 and 3:

my comments: Only because many studies were done in solution (with often truncated chimeric protein to induce solubility into membrane proteins) it does not mean one should ignore the fact that those proteins are anchored or even embedded in the mitochondrial outer membrane. And running solution NMR on their protein in membrane mimicking micelles might easily give them some ideas if their solution view is

reflected in a more native-like environment. And titration with compound 35 one could try using small amounts of DMS e.g. like many people do.

We did not ignore the fact that Bcl-2 is membrane anchored. The entire second paragraph of the discussion (pages 12-13) is dedicated to discuss potential issues arising from participation of the juxtamembrane region on binding, and mentions our unsuccessful attempts to study full-length Bcl-2 in nanodiscs because of the ill behavior of the protein in the presence of detergents, which need to be used for incorporation into nanodiscs. We did not attempt to work in detergent micelles, as suggested by the reviewer, because of this poor behavior of Bcl-2 and because the previously published spectrum obtained in DPC micelles (Fig. 4 of ref. 5 mentioned in the previous round of review) suggests that the protein becomes denatured in these micelles. We believe that it is not necessary to point this fact in the paper.

Reviewer #2 (Remarks to the Author):

The authors have responded to my concerns with additional discussion and some additional in silico analyses.

I have only one comment. The idea that 35 is a proof of concept that the Bcl-2-Bcln1 interaction can be disrupted to induce autophagy without inducing apoptosis is no longer supportable, given that other BH3's bind to Bcl-2 similarly to Beclin, and no additional data are provided regarding the effect of 35 on such binding. I suggest that this may still be a possibility, but the data do not strongly support the idea either way.

We never claimed that '35 is a proof of concept' and we still do not make this claim in the revised manuscript. We have been careful throughout the manuscript not to draw strong conclusions about the notion that the Bcl-2-Bcln1 interaction can be selectively disrupted, and whenever we present this notion we use soft terms like 'suggest', 'possibility' and 'potential'. We point out the problems that need to be addressed to develop selective inhibitors in the second to last paragraph of the discussion. We realize that a limitation of our work is that we have not tested the biological activity of 35 in cells and, as explained above in the responses to reviewer 1, we have included an explicit statement of this limitation in the revised manuscript.

We hope that we have properly addressed the reviewer concerns and that the paper can be accepted for publication in its current form. Please do not hesitate to contact me if you have any questions. Thanks again for handling our manuscript.

Sincerely,

Josep Rizo
UT Southwestern Medical Center
e-mail: Jose.Rizo-Rey@UTSouthwestern.edu
Phone: 214-645-6360